# Cardiac Defects and Genetic Syndromes: Old Uncertainties and New Insights

**DOI:** 10.3390/genes12071047

**Published:** 2021-07-08

**Authors:** Giulio Calcagni, Flaminia Pugnaloni, Maria Cristina Digilio, Marta Unolt, Carolina Putotto, Marcello Niceta, Anwar Baban, Francesca Piceci Sparascio, Fabrizio Drago, Alessandro De Luca, Marco Tartaglia, Bruno Marino, Paolo Versacci

**Affiliations:** 1Department of Pediatric Cardiology and Cardiac Surgery, Ospedale Pediatrico Bambino Gesù, IRCCS, 00165 Rome, Italy; marta.unolt@opbg.net (M.U.); anwar.baban@opbg.net (A.B.); fabrizio.drago@opbg.net (F.D.); 2Department of Pediatrics, Obstetrics and Gynecology, “Sapienza” University, 00161 Rome, Italy; flaminia.pugnaloni@gmail.com (F.P.); carolina.putotto@uniroma1.it (C.P.); bruno.marino@uniroma1.it (B.M.); paolo.versacci@uniroma1.it (P.V.); 3Genetics and Rare Diseases Research Division, Ospedale Pediatrico Bambino Gesù, IRCCS, 00165 Rome, Italy; mcristina.digilio@opbg.net (M.C.D.); marcello.niceta@opbg.net (M.N.); marco.tartaglia@opbg.net (M.T.); 4Medical Genetics Division, Fondazione IRCCS Casa Sollievo della Sofferenza, 71013 San Giovanni Rotondo, Italy; francesca.piceci@uniroma1.it (F.P.S.); a.deluca@css-mendel.it (A.D.L.)

**Keywords:** del22q11 deletion syndrome, down syndrome, Ellis–Van Creveld syndrome, congenital heart disease, genetic syndrome

## Abstract

Recent advances in understanding the genetic causes and anatomic subtypes of cardiac defects have revealed new links between genetic etiology, pathogenetic mechanisms and cardiac phenotypes. Although the same genetic background can result in different cardiac phenotypes, and similar phenotypes can be caused by different genetic causes, researchers’ effort to identify specific genotype–phenotype correlations remains crucial. In this review, we report on recent advances in the cardiac pathogenesis of three genetic diseases: Down syndrome, del22q11.2 deletion syndrome and Ellis–Van Creveld syndrome. In these conditions, the frequent and specific association with congenital heart defects and the recent characterization of the underlying molecular events contributing to pathogenesis provide significant examples of genotype–phenotype correlations. Defining these correlations is expected to improve diagnosis and patient stratification, and it has relevant implications for patient management and potential therapeutic options.

## 1. Introduction

Ongoing advances in molecular genetics are leading to an increasingly better understanding of the etiology of congenital heart diseases (CHDs). In the past, the specific role of genetic factors in the pathogenesis of these defects was not completely appreciated. Except for a few cases, it was thought that the most common cardiac lesions in the general population were those occurring in patients with chromosomal abnormalities, such as Down syndrome (DS) [1].

Occurrence of specific associations between CHDs and extracardiac (EC) malformations, including those defining genetic syndromes, has been reported within a high percentage of children with an atrioventricular canal defect (AVCD) (approximately 75%) and, less frequently, in children with a ventricular septal defect (VSD), patent ductus arteriosus (PDA) and Tetralogy of Fallot (ToF) (about 25%). Extremely rare associations have been reported when considering other cardiac defects, such as transposition of the great arteries (TGA) and pulmonary atresia (PA) with intact interventricular septum [2].

In the mid 1980s, a very important epidemiological study conducted in a large CHD population, the Baltimore–Washington Infant Study, highlighted the strong impact of genetic factors in the pathogenesis of CHDs, indicating that about 25% of children with CHD had associated EC malformations and that about one third of these presented an underlying known genetic syndrome, including chromosomal abnormalities and mendelian disorders or in the setting of an unrecognized syndromic condition [3].

More recent studies have confirmed these data, highlighting that EC anomalies are more frequent in patients with severe forms of CHDs and that the most commonly associated findings are skeletal malformations, gastrointestinal and genitourinary systems anomalies and neurodevelopmental disorders [4,5].

In 1977, de la Cruz demonstrated, through in vivo experimental studies on chicks’ embryos, that both the outflow and inlet tracts of the ventricles originate outside the heart. This finding emphasized the important contribution of EC tissues in the embryogenesis of the heart, justifying the strong association between the presence of CHD and EC malformations in the context of syndromic diseases [6].

In the last 20 years, studies have confirmed that the majority of the vertebrate heart is generated by the progressive addition of second heart field (SHF) progenitor cells, located in the splanchnic pharyngeal mesoderm, to the poles of the heart tube that elongates during cardiac looping. In this way, the SHF gives rise to the myocardium of the outflow tracts, the atria and the entrance to the ventricles [7,8,9]. In heart development, the critical contribution of precursors originating outside the primitive cardiac tube may explain the frequent co-occurrence of CHDs and EC anomalies in patients with genetic syndromes. Several genotype–phenotype correlation studies have suggested that, in patients with genetic syndromes, specific morphogenetic mechanisms induced and modulated by gene networks may result in specific cardiac phenotypes [10,11]. Therefore, well-defined cardiac anatomic subtypes may suggest accurate genetic diagnoses, which can be confirmed by molecular testing. Based on these studies, over the years, specific genotype–cardiac phenotype correlations have been accurately described in several genetic conditions, including DS [12,13,14], 22q11.2 deletion syndrome (22q11.2DS) [15,16,17,18], Noonan syndrome (NS) [19], Ellis–Van Creveld syndrome (EVCS) [20], Williams syndrome [21,22], 1p36 deletion syndrome [23] and Holt-Oram syndrome [24,25].

By now, it is known that underlying genetic conditions have an increasingly recognized impact on the anatomical and functional complexity of CHDs and could represent additional risk factors or even protective factors for cardiac surgery and clinical outcomes [26,27,28,29,30].

It is even more clear that early identification of the genetic causes of CHDs is expected to allow our understanding of the underlying pathogenetic mechanisms that influence the clinical and surgical outcome of a specific subtype of CHD [31]. Similarly, the identification of genotype–phenotype correlations is predicted to guide a more effective personalized management of patients with cardiac defects, improving quality of life and long-term outcome [30]. Indeed, recent studies showed that precision medicine using genotype–phenotype correlation data is able to guide not only risk stratification, but also identification of treatments that can modify the molecular mechanism of the disease [32,33,34].

In this review, we have described three different models of genetic syndromes associated with well-defined types of CHDs, showing how a specific genetic disorder can contribute to the expression of a specific cardiac phenotype. In particular, we have analyzed DS, 22q11.2DS and EVCS as examples of chromosomal aneuploidy, microdeletion abnormality and monogenic disorder, respectively.

## 2. Down Syndrome

Down syndrome (DS) (MIM # 190685) is the most frequent chromosomal syndrome, caused by an extra full or partial copy of chromosome 21 and is characterized by developmental delay, CHDs, gastrointestinal malformations and facial anomalies. The prevalence of CHDs in DS is about 50%, and anatomical types include AVCD, atrial septal defects (ASD), VSD and ToF [14]. The prevalence of CHD is lower in patients with mosaicism [35].

Specific anatomic types of CHD are diagnosed in children with DS. In fact, AVCD is prevalently “simple type” complete, and it is rarely associated with other CHDs, with the exception of ToF [35,36]. The comparison between patients with DS and those with AVCD and normal chromosomes has shown that associated CHDs, in particular left-sided obstructions, are significantly rarer in patients with DS [12,14,37,38]. The ‘inlet type’ VSD is prevalent in patients with DS, often associated with cleft of the mitral valve, whereas muscular and subarterial VSD are very rare [39]. In addition, the AVCD associated with ToF in DS is generally complete and type C of the Rastelli classification.

Clinical studies have shown that patients with DS have a better surgical prognosis in comparison to that found in children with CHD and normal karyotype, with a minor degree of anatomic variability [27,40]. Nevertheless, the role of noncardiac factors should be considered as negative prognostic factors in the surgical outcome, particularly immune deficits, pulmonary infections and hypertension due to increased vascular resistance [27].

Atypical CHDs in DS patients can be found in a minority of patients, including left-sided lesions, structural myocardial changes and univentricular physiology [41]. In this subgroup of patients, a significant excess of multiple surgeries is documented, in consideration of the multiple and complex anatomical defects. Univentricular palliation represents a challenge in prognostic terms, since it is very rare that a patient with DS can reach Fontan stage palliation. In addition, specific minor morphological cardiac characteristics have been described in patients with DS without CHDs, including enlargement of membranous septum and abnormal offsetting of the atrioventricular valves with a tendency toward the insertion at the same level [42]. These subclinical characteristics have been confirmed more recently also in fetuses with DS, with dysplasia of atrioventricular valves as an additional sign [43].

The involvement of genetic factors in determining epidemiological and anatomical differences in patients with DS is supported by clinical and molecular evidence. In fact, ethnic differences in the prevalence of specific types of CHDs have been noted, since ASD and VSD are the CHDs more frequently diagnosed in oriental and native American patients with DS [44], while the first CHD for frequency in Caucasian children with trisomy 21 is AVCD [45].

It can be hypothesized that the overdosage of genes mapping on chromosome 21 can lead to specific congenital malformations. Genes located on chromosome 21 which are possibly involved in the pathogenesis of AVCD in DS include *DSCAM*, *COL6A1*, *COL6A2* and *DSCR1* [46,47]. Particularly, *DSCAM* encodes a cell adhesion molecule and is expressed in the heart during cardiac development. Collagen VI is expressed in the basement membranes of endothelia and in the endocardial cushions during development. Recent case–control genome-wide association studies have identified some CHD risks loci located within a so-called ‘CHD critical region’ on chromosome 21, one mapping near *RIPK4* and the second one including *ZBTB21* [48]. The segment residing 6 kb upstream of RIPK4 may contain regulatory elements for this gene, and overexpression of RIPK4 has been demonstrated in the heart tissue of the Ts65Dn Down syndrome mouse model [49]. The *ZBTB21* gene has been shown to interact with *PPP2R2B*, which, in Drosophila, regulates the WNT/β-catenin signaling pathway. This pathway is required for cardiac differentiation in human embryonic stem cells. Additional analysis of copy number variants has also been performed, showing no single pathogenetic microrearrangement on chromosome 21 in DS patients with AVCD [50]. In general, a multifactorial model for the development of CHD in DS is suggested, with a cumulative effect in multiple risk alleles having a minor or major contribution.

The role of genetic modifiers mapping outside chromosome 21 in clinical expression of CHD in DS is corroborated by studies in humans and experimental mice. In fact, first studies documented variants in *CRELD1, CRELD2*, *GATA4*, *GATA5*, *FBLN2, FRZB* and *BMP4* as AVCD susceptibility factors [46,47]. Additionally, a high prevalence of the c.973G > A (p.Glu325Lys) variant in *CRELD1* has been detected recently in patients with DS and AVCD [51]. Interestingly, several microRNAs (miRNAs) mapping not only on chromosome 21, but also on other loci have been reported to be overexpressed in DS, including miR-155, miR-802, miR-125b-2, let-7c, miR-99a and the miR-99a/let-7c cluster [52,53]. Studies on mouse models are also important, since they can be related to humans. Mouse models of DS consisting in crossing loss-of-function alleles of *Creld1* or *Hey2* genes onto the trisomic background showed a higher frequency of CHD [54]. Studies generating mouse strains with segmental duplications identified two segments, including seven coding genes (Setd4, Mx2, Tmprss2, Ripk4, Prdm15, C2cd2 and Zbtb21) [55]. Furthermore, the involvement of the Jam2 gene in Ts65Dn mouse as a potentiator of the genetic modifier Creld1 has been identified [56]. Additionally, in DS and in additional genetic syndromes with AVCD, a link to the sonic hedgehog signaling has been documented [57,58]. Particularly, studies on cerebral, skin, liver and intestine mice trisomic cells have shown a defective mitogenic hedgehog activity with cell proliferation impairment due to a higher expression of Ptch1, a receptor normally repressing the hedgehog pathway [57].

In conclusion, specific anatomic cardiac characteristics have been documented in DS, with ongoing molecular and experimental studies documenting a large heterogeneity of genetic predisposing factors not only within the “extracopy of chromosome 21”, but on a global genomic level.

## 3. 22q11.2 Deletion Syndrome

22q11.2DS (MIM # 192430 #188400) is the most frequent chromosomal microdeletion disorder, found in 1/4000 live births and 1/1000 fetuses. More than 85% of affected individuals carry a 3 megabase hemizygous deletion between LCR22A and LCR22D. However, smaller and nested proximal or distal deletions are present in some individuals with 22q11.2DS [59]. The dimension of the microdeletion seems not to influence the phenotype, while the localization (proximal versus distal) is related to a slightly different prevalence of the associated features [60]. In patients with the 3 Mb deletion and those with the proximal nested deletions, including the LCR22A–LCR22B region, the prevalence of CHD is approximately 65%. A lower prevalence (∼32%) is observed in individuals with distal nested deletions [17,61].

As described by the studies published in the 1990s, most 22q11.2DS patients with CHD have conotruncal heart defects (CTDs), due to an impaired development of the cardiac outflow tract, including the aortic arch. Such defects include TOF and PA in 10–25% of cases. Persistent truncus arteriosus (PTA), interrupted aortic arch type B (IAAB) or right-sided aortic arch (RAA) with abnormal branching of the subclavian arteries have been reported in 35%, 50% and less than 5%, respectively. Some have isolated VSD, atrial septal defects (ASD) and, rarely, other CHDs [15,60,62,63,64].

While genotype–phenotype correlations between the presence of the microdeletion and the risk of CHD is well-established, the pathogenetic and embryogenetic pathways leading to this strong correlation is still yet to be completely understood. One of the main unsolved questions remains the high phenotypical variability of the syndrome: in some patients, the typical cardiac, palatal and facial features are absent, and the clinical presentation includes some high-prevalence conditions associated with 22q11.2DS, such as hypocalcemia or psychotic illness. In these cases, the diagnosis may be missed. This clinical variability is also intrafamilial, with patients being diagnosed only following the birth of an affected child or sibling [65,66].

So far, the high risk of having a CHD in 22q11.2DS has been mainly attributed to the presence of the hemizygous deletion. Thus, many studies have been focused on the role of the proteins encoded by genes located within the DiGeorge critical region (DGCR). Among these, *TBX1* has been demonstrated as a key transcription factor, based on multiple mouse model approaches [67,68,69]. Indeed, some authors have reported mutations in TBX1 in unrelated patients who did not have a 22q11.2 deletion but had the same clinical phenotype [70,71]. However, the high phenotypic variability characterizing the syndrome cannot be fully explained only by the contribution of the 22q11.2 deletion or the size of the deleted region, supporting the idea that additional genetic and/or environmental factors act as modifiers [17,60].

In particular, a very recent paper provided new interesting insights into genetic modifiers of CHDs in 22q11.2DS [72]. Zhao et al. tested the hypothesis that common and/or rare single-nucleotide variants on the remaining 22q11.2 allele might be specifically associated with CHDs. The authors found that common variants located in a 350 kb region within the LCR22C–LCR22D intervals on the retained allele were significantly associated with an increased risk for CTDs in individuals with the typical 3 Mb 22q11.2 deletion. These variants were shown to lie within regulatory regions of *CRKL*, one of the four genes contained in this region. Mouse model data already suggested that *Crkl* is important in neural-crest cells within the pharyngeal apparatus for cardiovascular development, as inactivation of both alleles of *Tbx1* or *Crkl* in mice results in a similar spectrum and range of CTDs. Therefore, the authors concluded that variability in CTDs penetrance in the 22q11.2DS population can be explained in part by variants affecting *CRKL* expression.

Further studies are needed in order to explore other hypotheses and models. Deeper understanding of the pathogenetic pathways leading to the presence of CHDs in patients with 22q11.2DS may also help in understanding the cause of coexisting comorbidities that may influence the clinical outcome of patients with both 22q11.2DS and their associated CHDs.

Since the first report on cardiac outcomes in patients with 22q11.2DS revealed a higher mortality rate, particularly after cardiac surgery, nowadays, with the increase of medical and surgical supports, the post-surgical outcome in these patients has been significantly improved [73,74]. However, an increased mortality rate, particularly in the subgroup of patients with ToF and PA-VSD, has still been reported by different authors. Probably, a role of a specific worse anatomic pattern in patients with CTD and 22q11.2DS, such as hypoplastic pulmonary arteries, has been advocated for its possible role in determining this outcome. Anaclerio et al. [26] demonstrated a higher mortality in 22q11.2DS patients compared to not-deleted ones when ToF and PA-VSD occurred. Different authors more recently confirmed these previous reports, particularly in the subgroup of 22q11.2DS patients with ToF plus PA-VSD and aorto-pulmonary collaterals [75].

A significant increase of peri/post-surgical complications have been reported in these patients. Specifically, cardiopulmonary complications, such as heart failure, respiratory failure, longer intubation/tracheostomy, laryngeal stridor and pneumonia and sepsis, have been advocated as responsible for this worse outcome [76]. Recently, Mercer-Rosa and colleagues analyzed data on the post-surgical course in patients with 22q11.2DS and ToF, demonstrating a higher prevalence of the Blalock–Taussig shunt prior to primary repair, longer and complicated intensive care management and a definitively longer hospital stay in patients with 22q11.2DS compared to not syndromic ones [77].

According to this evidence, clinical management of cardiac patients with 22q11.2DS seems to be a very important step, and appropriate counseling and anticipatory care should also be provided. Genetic testing from fetus to childhood, prior to any surgical steps, should be performed in order to prevent all complications known. Management of complex CHDs as in 22q11.2DS patients using standard protocols may not be enough in these children. Only being informed of the specific clinical needs of these patients allows for the improvement of a tailored clinical and surgical management according to the presence of the genetic condition.

## 4. Ellis–Van Creveld Syndrome

EVCS is an autosomal recessive skeletal dysplasia that occurs in 1 in 60,000 to 200,000 newborns. Together with Sensenbrenner syndrome (MIM# 218330), Jeune syndrome (MIM# 208500) and the subgroup of lethal short-rib-polydactyly syndromes (SRPS1, Saldino-Noonan syndrome, MIM# 613091; SPRS2, Majewski syndrome, MIM# 263520; SPRS3, Verma-Naumoff syndrome, MIM# 613091; SPRS4, Beemer-Langer syndrome, MIM# 269860), EVCS can be clumped within the group of skeletal ciliopathies [78], a family of clinically variable and genetically heterogeneous group of disorders due to genetic mutations that disrupt proper formation of cilia. These are organelles projecting from the cell membrane that are essential for several biological processes during development, including transduction of many signaling pathways. A distinctive feature that EVCS shares with other ciliopathies is postaxial polydactyly. Other ciliopathies with postaxial polydactyly include oral–facial–digital syndromes (OFD, MIM# 258860), Bardet–Biedl syndrome (BBS, MIM# 209900) and Smith–Lemli–Opitz syndrome (SLOS, MIM# 270400). Of note, in these conditions, postaxial polydactyly is frequently associated with AVCD. CHDs, AVCD in particular, represent a cardinal feature of EVCS, with almost 60% of patients being affected. Other phenotypic aspects of EVCS include short limbs, short ribs, postaxial polydactyly and dysplastic nails and teeth.

The striking association of single atrium and postaxial polydactyly was firstly described by Levin et al. and Onat et al. [79,80]. At the beginning of the 1990s, our group described three unrelated individuals showing a clinical association of polydactyly, single atrium and AVCD. All patients also showed systemic venous return anomalies (prevalently persistent left superior vena cava). Although the studied patients did not show signs of acromesomelic dwarfism, dental anomalies and oral frenula, the study of the phenotypic aspects of the patients drove our group to consider the hypothesis that EVCS and single atrium/polydactyly syndromes might be related, possibly due to allelic mutations [81]. Beginning from that study, our group has considered a specific cardiac phenotype (AVCD with venous return anomalies) as a cardinal feature of EVCS. Subsequently, the clinical observation that EVCS and the condition described as “single atrium, atrioventricular canal/postaxial polydactyly” could represent variable phenotypic expressions of the same disease was corroborated by further case reports described in 1997 [82]. A detailed description of the phenotypic spectrum of patients allowed us to consider that EVCS has a phenotypic overlap with other genetic conditions characterized by CHD and postaxial polydactyly, such as Kaufman–McKusick syndrome (MKKS), leading us to raise the hypothesis that a spectrum of phenotypic expressions, varying from AVCD/polydactyly syndrome to more complex malformation associations, does exist.

The careful study of anatomic types of CHDs can reveal the pathogenetic relation linking different syndromes, indicating a correct definition of genotype–phenotype correlations. Hypothesizing a common pathogenetic relation underlying their clinical overlap, AVCD was thus considered a crucial phenotypic aspect linking EVCS to other skeletal ciliopathies and OFD syndromes [82,83]. The phenotypic similarities detected by a detailed description of cardiac anatomy found their molecular turning point in the study of Ruiz Perez and colleagues, who isolated, by positional cloning, the *EVC* gene mutated in most EVCS patients [84]. A few years later, the same authors identified the second gene causative of EVCS, named *EVC2*, which is located close to *EVC* on chromosome 4 in a divergent orientation [85]. Of note, our group recently identified hypomorphic mutations in the *EVC* gene in patients with AVCD, common atrium and postaxial polydactyly, confirming that, at least in some cases, this association represents a mild clinical subtype of EVCS [86].

In situ hybridization and immunofluorescence studies were later used to study *EVC* and *EVC2* genes’ function and expression profiles in mouse tissues and whole embryos. Sund and collegues found colocalization of *Evc* and *Evc2* mRNA and protein in developing mouse hearts. In particular, the gene expression profile was more evident in the SHF (both outflow tract and dorsal mesenchymal protrusion), but also in mesenchymal-derived cardiac tissues, like the atrial septum and the atrioventricular cushions, suggesting the molecular explanation of the peculiar cardiac phenotype observed in EVCS patients [87]. *EVC* and *EVC2* encode for transmembrane proteins that form a protein complex, transducing extracellular signals to the nucleus via the hedgehog signaling pathway [88]. Loss of function of *EVC* or *EVC2* leads to absence of the protein complex so that when hedgehog signaling is activated, SMO cannot lead to full activation of GLI, which is a direct hedgehog target. As a result, the cilia-mediated response to hedgehog ligands is diminished [89].

The LOVD database reports 133 pathogenic variants in EVCS patients, 70 in the *EVC* gene and 63 in the *EVC2* gene (https://databases.lovd.nl, July 6^th^ 2021). The majority of *EVC* and *EVC2* mutations introduce premature termination codons, either directly or following a frameshift, thus predicting that they are loss of function mutations. Mutations in both genes include single-base pair substitutions, microinsertions, microdeletions and few deletions spanning more than one exon within the same gene [88].

In addition to clinical variability linked to the allelic heterogeneity of the *EVC* and *EVC2*, molecular analysis of EVCS cases have suggested a more complex genetic heterogeneity. In particular, specific mutations in the *WRD35* gene, the gene most commonly mutated in Sensenbrenner syndrome, have been found in rare patients showing a particular EVCS phenotype [89]. *WRD35* plays a role in hedgehog signaling, since this gene encodes for IFT121, which is part of the IFT-A subunit, which interacts with IFT-B in a complex responsible for circulation of protein cargos along the cilium. Immunofluorescence analysis of *Wdr35*-/- mouse fibroblasts demonstrated that IFTA is needed for correct ciliary entry of EVC, EVC2 and SMO [89]. Further expanding EVC genetic heterogeneity, *DYNC2LI1* variants were subsequently found by our group in several pedigrees showing EVCS-like phenotypes [90]. *DYNC2LI1* encodes for a protein of the dynein-2 complex that interacts with WDR35 to drive retrograde circulation of cargos in the cilium [91]. Moreover, additional phenotypes overlapping with EVCS have been associated with biallelic truncating mutations in *GLI1,* leading to protein inactivation and hedgehog signaling impairment [92]. More recently, germline and mosaic variants in *PRKACA* and *PRKACB* were identified in patients affected by an EVCS-like multiple congenital malformation syndrome [93]. *PRKACA* and *PRKACB* encode for two catalytic subunits (Cα and Cβ) of cAMP-dependent protein kinase (PKA), a pleiotropic holoenzyme that regulates numerous fundamental biological processes, such as metabolism, development, memory and immune response. According to the central role of hedgehog signaling impairment in EVCS-related phenotypes, PKA works to restrain hedgehog signaling through phosphorylation of GLI transcription factors [94]. EVCS disease genes and their prevalence are shown in Table 1.

## 5. Conclusions

Recent advances in cytogenetic and molecular diagnostic technologies have greatly improved the ability to identify new genes and chromosomal regions involved in syndromic and non-syndromic CHDs, allowing more precise definition of the pathogenetic mechanisms underlying these defects. Although the complex and multifactorial mechanisms of inheritance still make the knowledge of the genetic causes of non-syndromic CHDs very limited, a deeper understanding of the genetic background of syndromic CHDs is expected to allow an increasingly detailed characterization of the genetic basis of isolated presentation of these defects.

Precise and early genetic diagnosis of patients with CHDs and genetic syndromes will permit a more personalized management, following multidisciplinary protocols and specific guidelines, and it provides the opportunity for a more effective genetic counseling. A better comprehension of the genetic factors related to CHDs will also lead to the implementation of new preventive strategies and targeted therapies for the needs of each individual, improving clinical, surgical management, long term outcome and quality of life of these patients.

## Figures and Tables

**Table 1 genes-12-01047-t001:** Genetic heterogeneity of Ellis–van Creveld syndrome.

Gene	Chromosome Location	OMIM	Prevalence
*EVC*	4p16.2	604831	31–62%
*EVC2*	4p16.2	607261	21–38%
*WDR35*	2p24.1	613602	<1%
*DYNC2LI1*	2p21	617083	~2%
*GLI1*	12q13.3	165220	<1%
*PRKACA*	19p13.12	601639	<1%
*PRKACB*	1p31.1	176892	<1%

## Data Availability

Not applicable.

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
