# Peer review of "Cardiac Defects and Genetic Syndromes: Old Uncertainties and New Insights"

_genes, 2021, doi:10.3390/genes12071047_

Round 1
Reviewer 1 Report
The manuscript of Calcagni et al reviews three well-known genetic syndromes, the link with CHD and the most recent advances herein. It’s a clear manuscript and I only have a few points.
In the DS section there the authors talk about genes outside chr21 contributing to the phenotype. The authors also briefly mention mouse models of DS here. I would like more discussion and information on these DS mouse models, as these mouse models of DS, or forms of DS, might also contribute to understanding which genes are involved. This could then be related back to human or lab data.
In the 22q11 section the authors do discuss the phenotypic variability. I would also like them to point out that many adult patients with 22q11 deletions are not recognized as such due to its phenotypic variability, and therefore not diagnosed as 22q11. This has important personal and clinical consequences for these patients.
In the EVC section a bit more background on the nature of EVC mutations and whether or not EVC is seen as loss or gain of function would be welcome, especially as the genetic architecture here is very different than the two chromosomal syndromes discussed previously.
Reviewer 2 Report
One overall comment: 1) maps/gene lists of the Chr22qdel, Ellis-VanCrefeld and Trisomy 21 regions would be helpful for the general reader.
2) Discussion of the incidence/prevalence of these genetic variants in the general population should be provided. (Note: words such as rare and common should be defined with actual frequencies)
- a) Chr22qdel:
a.1) The authors state: "Among 185 these, TBX1 has been demonstrated as key transcription factor, based on multiple mouse 186 model approaches [62-64]. "
If TBX1 is the 'key transcription factor' then humans with TBX1 mutations should also have clinical features of chr22qdel. What is the state of the literature regarding TBX1 mutations and human disease?
- b) Trisomy 21: Discussion of critical region genes would be useful
Minor/major?
The concluding sentence:
A closer cooperation among pediatric car-339 diologists, clinical and molecular genetists may allow to improve our knowledgment, 340 long term outcome and quality of life of patient. 341
342
requires editing.
Round 2
Reviewer 2 Report
no further concerns.